# Qualitative and Quantitative Analysis of Edible Bird’s Nest Based on Peptide Markers by LC-QTOF-MS/MS

**DOI:** 10.3390/molecules27092945

**Published:** 2022-05-05

**Authors:** Wen-Jie Wu, Li-Feng Li, Hui-Yuan Cheng, Hau-Yee Fung, Hau-Yee Kong, Tin-Long Wong, Quan-Wei Zhang, Man Liu, Wan-Rong Bao, Chu-Ying Huo, Quan-Bin Han

**Affiliations:** 1School of Chinese Medicine, Hong Kong Baptist University, 7 Baptist University Road, Kowloon Tong, Hong Kong 999077, China; 18482767@life.hkbu.edu.hk (W.-J.W.); 16483294@life.hkbu.edu.hk (L.-F.L.); hycheng10@163.com (H.-Y.C.); 11018860@life.hkbu.edu.hk (H.-Y.F.); 16223551@life.hkbu.edu.hk (H.-Y.K.); 15485021@life.hkbu.edu.hk (T.-L.W.); 18482422@life.hkbu.edu.hk (Q.-W.Z.); liuman@hkbu.edu.hk (M.L.); 16483502@life.hkbu.edu.hk (W.-R.B.); 20481969@life.hkbu.edu.hk (C.-Y.H.); 2Hong Kong Authentication Centre of Valuable Chinese Medicines, Hong Kong 999077, China

**Keywords:** edible bird’s nest, peptide markers, qualitative and quantitative analysis

## Abstract

Edible bird’s nest (EBN) is an expensive health food. There are many adulterants in the market. It remains challenging to discriminate EBN from its adulterants due to a lack of high-specificity markers. Besides, the current markers are confined to soluble fraction of EBN. Here, both soluble and insoluble fractions were analyzed by LC-QTOF-MS/MS. A total of 26 high-specificity peptides that were specific to EBN were selected as qualitative authentication markers. Among them, 10 markers can discriminate EBN from common adulterants, 13 markers discriminate white EBN from grass EBN/common adulterants, and 3 markers discriminate grass EBN from white EBN/common adulterants. Three of them, which showed high signal abundance (Peak area ≥ 10^6^) and satisfactory linearity (*R*^2^ ≥ 0.995) with EBN references, were selected as the assay marker; and their peptide sequences were confidently identified by searching database/de novo sequencing. Based on these markers, a qualitative and quantitative analytical method was successfully developed and well-validated in terms of linearity, precision, repeatability, and accuracy. The method was subsequently applied to detect EBN products on the market. The results indicated that more than half of EBN products were not consistent with what the merchants claimed.

## 1. Introduction

Edible bird’s nest (EBN, Yan wo in Chinese), created by the swiftlet, is an expensive animal-derived product from Southeast Asia. Proteins are the major components in EBN, accounting for 42–63% of its dry mass [1,2]. Traditionally, EBN is served as a food tonic and is believed to have high health benefits [3].

With the growth of market demand, there is a lot of news reporting fake EBN or the selling of low-quality EBN at a high price [4]. In the market, according to the amount of impurities, EBN is classified into three types: white, feather, and grass nest. It is widely acknowledged that the white nest is of a higher quality, so its price is several-fold higher than that of the grass nest [5,6,7]. Some much cheaper adulterants are also found in the market. Among the most common adulterates, agar and tremella fungus mainly contain polysaccharides, while gelatin and pork skins are mostly composed of polypeptides [8,9,10,11]. The adulterants always have a similar appearance and taste (Figure 1). Hence, it is hard for people to distinguish them just by observing their appearance. Furthermore, the addition of adulterants to the instant EBN products may reduce the health benefits of EBN or even induce health hazards. In addition, the grass nest is popularly used to simulate the white nest in these products. These issues show the importance of the authentication of EBN. Otherwise, the market will remain in chaos, and consumers’ health will be at risk.

The currently-used authentication methods have their own weaknesses. Sialic acid is a widely used authentication marker; however, it is not specific enough and can also be found in many other foods such as eggs, red meat, or dairy products [12,13,14]. For the protein analysis, real-time PCR (targeting the genes of fibrinogen and NADH dehydrogenase) and two-dimensional gel electrophoresis were used as the Chinese standard [15,16]. In other studies, amino acid and peptide fingerprints were employed [17,18,19]. However, these approaches have low specificity since not only EBN has these genes and proteins. Additionally, there are a lot of additives, resulting in the poor recognition of characteristics belonging to EBN itself. Recently, efforts on peptide markers have been made using shotgun proteomics combined with multiple reaction monitoring (MRM) analysis [20]. Although the specificity is improved, the use of this method is still limited in quality analysis because only the minor water-soluble protein is addressed while the insoluble protein is the majority. The protein information provided by this method is incomplete. Additionally, the method only compared four batches of EBN samples with four types of adulterants, and the specificity of markers has not yet been confirmed by large numbers of samples. More importantly, none of the methods mentioned above could distinguish the white from the grass nest.

In this study, we aim to find peptide markers for the authentication of EBN, especially white EBN. To this end, we developed a simple and effective method for sample treatment, which avoids losing potential markers during the process and is practicable to analyze large numbers of samples. Both soluble and insoluble fractions of EBN were analyzed to find peptide markers. Subsequently, a strategy without relying on protein databases was employed to find as many peptide markers as possible. We found 26 new specific markers. There was a good linear relationship between the peak area of three high-signal abundance markers and the total protein content in EBN/mass of EBN, which suggested that they were acceptable for quantitative analysis. Their peptide sequences were confidently identified by searching database/de novo sequencing. These new markers were successfully applied to detect EBN content in commercial products.

## 2. Results

### 2.1. Selection of EBN-Specific Peptides

In this study, the solution and residue of EBN were digested simultaneously, then formic acid was added to make the residue soluble (Appendix A). So, both the solution and residue were analyzed to find peptide markers. Using LC-QTOF-MS/MS, the peptide profiles of digested EBN and its common adulterants were obtained based on base peak chromatographs (BPCs) from a scan range of *m/z* 100–1200 (Figure 1). By comparing each high-sensitivity ion of digested white EBN with its common adulterants including agar, chicken egg white, gelatin, milk powder, pork skin gelatin, tremella fungus, rice, starch, and swim bladder, 23 specific peptide markers were found (Table 1). The specificity of these markers was further confirmed in multiple batches of EBN and its common adulterants based on the extracted ion chromatogram (EIC).

As shown in Table 1 and Appendix A, these markers should satisfy three criteria: (1) the peak should be found only in digested EBN, but not in any of the adulterants; (2) the peak should be a peptide; (3) the peak should be consistently found in multiple batches of EBN. There were 10 peptides specific to EBN based on the marker selection criteria. These peptides were consistently presented in the multiple collected EBN (overlapped chromatograms) samples (*n* = 70). Thus, these peptides are satisfactory as EBN authentication markers. Moreover, based on the criteria of marker selection, 13 peptides are specific to white EBN because they have consistently presented in the multiple batches of white EBN (overlapped chromatograms) samples (*n* = 60). Similarly, three peptides were specific to grass EBN (*n* = 10). The information is summarized in Table 1.

### 2.2. Identification of Specific Peptides

To identify the sequence of these marker candidates, the MS/MS data were used for searching the database and de novo sequencing. As shown in Table 2, three marker peptides were confidently identified. The sequence was AMESINSR, VSAPGPVLTR and SDDSLWR, respectively. As shown in Figure 2, the corresponding b/y ions in the MS/MS spectra were provided, suggesting the marker peptide identification was reliable. The remaining markers failed to be identified or confidently identified. This may be caused by the limited protein information of EBN in the database. As searched in the UniprotKB, only 4025 protein items were for Apodidae and 104 protein items for *Aerodramus fuciphagus* (edible-nest swiftlet).

Furthermore, the EICs of EBN and its common adulterants were compared to confirm the specificity of these identified markers. As shown in Figure 3, these peptides were of a high specificity. Using these markers, the white EBN and grass EBN could be well-discriminated.

### 2.3. Quantitative Analysis Based on EBN-Specific Markers

For quantitative analysis, a marker should satisfy three criteria: (1) the peptide sequence is confidently identified (Figure 2); (2) the marker is of high signal abundance (peak area > 10^6^); (3) the marker shows satisfactory linearity (*R*^2^ ≥ 0.995), precision, repeatability, and accuracy (Table 3 and Figure 4). Fortunately, the three identified marker peptides satisfy these three criteria, so the EBN quantitative analysis was applied to 46 batches of liquid products collected from China and Vietnam. The retention time and mass to charge (*m/z*) were used to confirm the existence of three marker peptides. The results are shown in Table 4 and Figure 5. The results showed that about 21 batches of EBN products had no EBN-related marker peptides. The findings indicate that there may be no or a very low amount of EBN in these claimed EBN products. This suggests that development of an effective EBN authentication method is quite meaningful to both customers and the industry. Additionally, 11 batches of products had both white and grass EBN markers. Furthermore, 14 batches of products were found to be made only from white EBN. These results indicate that the misusage of white and grass EBN, especially in instant EBN products, is very widespread.

## 3. Discussion

Numerous studies reported chemical components in the soluble extract of EBN [21,22,23]. However, according to the traditional method of EBN preparation and consumption, people eat the entire expensive delicacy, including both the soup and the residue; however, the components in the residue have not been considered in previous studies. Therefore, in this study, we tried to make samples fully soluble to assess the quality of EBN objectively. To this end, 0.1 M HCl (pH = 1) was first used as it was reported to help the [24] insoluble fraction readily digested by the pepsin (pH at 0.5–1.5). But in this study, adding 0.1 M HCl failed to make samples fully soluble. In addition, stewing the samples for a long time was also employed. Unfortunately, there was still some insoluble matter upon cooling to room temperature. After centrifugation, the supernatant (soluble fraction) or residue (insoluble fraction) was placed at −80 °C refrigeration overnight and lyophilized until completely dried. The average protein content in the soluble fraction (BCA Protein Assay) was only about 8%. So, chemical components in the insoluble fraction should not be ignored. Eventually, we found that samples became acid-soluble after digestion by trypsin (Appendix A). Therefore, EBN raw materials were directly digested by trypsin in this study. After digestion, the samples were fully dissolved by adding a certain amount of formic acid. Furthermore, we found that the insoluble fraction even had bigger peak areas of peptide markers than the soluble fraction (Appendix A), which suggests it is extremely important to make samples fully soluble.

To discover the specific peptide markers, the shotgun proteomic analysis is always employed. However, it will have some limitations when applied to EBN. For example, as most of the proteins in EBN are glycosylated with O and/or N-glycans, modification increases the difficulty of protein separation, analysis, and identification. Thus, complex pretreatment including glycan removal is always necessary. Simultaneously, the reduction and alkylation steps are also involved in discovering more protein information. These pretreatments will be more time-consuming and not practicable for commercial testing. Therefore, to make the authentication of EBN more simple and easy-to-operate, direct trypsin digestion and comparison were used in this study.

Interestingly, the quantitative analysis results showed that the content calculated by E7 was not the sum of W2 and G1. Besides, the contents calculated by W2 and E7 were very close. The cause may be that (1) E7 was mainly contributed by white EBN, while the contribution from grass EBN was minimal; (2) the same reference was used to build the linearity of W2 and E7, while no mixed reference was used to establish linearity of W2, E7 and G1 simultaneously; (3) the relative contents of W2 and E7 were not fixed in different samples.

## 4. Materials and Methods

### 4.1. Chemicals and Materials

Trypsin (sequencing grade) was bought from Shang-hai Yuanye Bio-Technology Co., Ltd. (Shanghai, China). Ammonium bicarbonate (NH_4_HCO_3_) and formic acid (FA) were purchased from Sigma Aldrich (St. Louis, MO, USA). LC-MS-grade acetonitrile and methanol were provided by RCI Labscan Limited (Bangkok, Thailand). Water used was purified with Millipore Milli-Q water purification system. The multiple batches of EBN (*n* = 70) including white EBN (*n* = 60) and grass EBN (*n* = 10) were collected from a local market (Hong Kong, China). The agar (*n* = 10), egg white (*n* = 5), gelatin (*n* = 8), cow milk (*n* = 8), pork skin (*n* = 4), tremella fungus (*n* = 6), rice flour (*n* = 4), starch (*n* = 3) and swim bladder (*n* = 3) were purchased from the market (Hong Kong, China). These samples were fully dried and then ground into powder. The edible bird’s nest related products (*n* = 46) were purchased from the market.

### 4.2. Sample Preparation

The powder (5 mg) of the EBN sample or adulterants was dissolved in 500 μL 1% ammonium bicarbonate (ABC) and sonicated for 30 min. For EBN products, 250 μL of product was diluted with 250 μL 2% ammonium bicarbonate (ABC). Then 50 μL trypsin (10 mg/mL in 1% ABC) was added. The mixture was incubated at 37 °C for 18 h. After digestion, the hydrolysate solution was acidified with 20 μL formic acid to ensure a pH of less than 4. Then 900 μL 1% ABC was added to dilute the samples. After that, 400 μL of the diluted sample was mixed with 1 mL methanol. The mixture was centrifugated at 15,000 rpm for 15 min. The supernatant was collected for further LC-QTOF-MS/MS analysis.

### 4.3. LC-QTOF-MS/MS Analysis

The separation was performed on an Agilent 1290 UHPLC system (Agilent Technologies, Santa Clara, CA, USA) equipped with a binary pump, a column oven, an auto-sampler, a degasser, and a diode-array detector. The system was controlled by MassHunter B.06 software. An ACQUITY UPLC BEH C18 (2.1 mm × 100 mm, 1.7 μm, Waters, Milford, CT, USA) chromatographic column was used and eluted with a linear gradient of 0.1% formic acid in water (A) and 0.1% formic acid in acetonitrile (B), at a flow rate of 0.35 mL/min and at a temperature of 40 °C. The solvent gradient programming was as follows: 0–5 min, 1% B; 5–30 min, 1–25% B; 30–32 min, 25–75% B; 32–33 min, 75–100% B; 33–34.1 min, 100–1% B; 34.1–38 min, 1% B. The injection volume was 2 μL (the amount of peptides is about 2 μg).

MS data were collected from an Agilent 6540 Q-TOF mass spectrometer (Agilent Technologies, Santa Clara, CA, USA) equipped with a quadrupole-time-of-flight (Q-TOF) mass spectrometer and a JetStream electrospray ion (ESI) source. Data acquisition was controlled by MassHunter B.06 software (Agilent Technologies). The optimized operating parameters in the positive ion mode were as follows: nebulizing gas (N2) flow rate, 7.0 L/min; nebulizing gas temperature, 300 °C; JetStream gas flow, 8 L/min; sheath gas temperature, 350 °C; nebulizer, 40 psi; capillary, 3000 V; skimmer, 65 V; Oct RFV, 600 V; and fragmentor voltage, 150 V. The mass scan range was set as mass to charge (*m/z*) 100–2000. MS/MS produces parallel alternating scans that provide precursor ion information at low collision energy, while MS/MS produces full scans that offer information about fragment masses, precursor ions and neutral loss at high collision energy. The collision energies for Auto MS/MS analysis were 20 V and 40 V.

### 4.4. Data Analysis

For searching the database, the raw data collected from Q-TOF-MS/MS were output as *mgf format using MassHunter B.06 Software. The *mgf files were loaded into ProteinPilot 5.0 software to match proteins and peptides. The corresponding *mgf data were searched against *Aerodramus fuciphagus* (104 protein items), *Apodidae* (4025 protein items) and Bird (94,241 protein items) databases downloaded from UniProt KB (29 June 2020). The parameters of ProteinPilot 5.0 were: Identification sample type; Iodoacetic acid cysteine alkylation; Trypsin digestion; Thorough search effort. An integrated false discovery rate (FDR) analysis was applied to establish a confidence level for the results.

For de novo sequencing, the raw data were submitted to de novo sequencing in PEAKS Studio software. The parameters for de novo sequencing were tolerance of precursor and fragment mass of 0.1 Da; De novo peptides, whose average local confidence (ALC) scores ≥95% were selected.

## 5. Conclusions

In summary, by systematically analyzing peptide profiles from tryptic digestion of EBN, we found 26 specific peptide markers including 10 EBN-specific, 13 white EBN-specific and 3 grass EBN-specific markers. The specificity of these markers was confirmed by large numbers of samples. Three of them showed satisfactory linearity, precision, repeatability, and accuracy, and their peptide sequences were confidently identified. Subsequently, a qualitative and quantitative analytical method was successfully developed and applied to detect EBN products on the market. This method revealed that only 14 out of 46 batches of commercial EBN products (products which claimed to include EBN) were credible, while the remaining products either did not contain EBN (21 batches), or contained grass EBN (11 batches). These results suggest that this method effectively improves the quality control of commercial EBN products, as it can not only discriminate EBN from common adulterants, but can also distinguish white EBN from grass EBN.

## Figures and Tables

**Figure 1 molecules-27-02945-f001:**
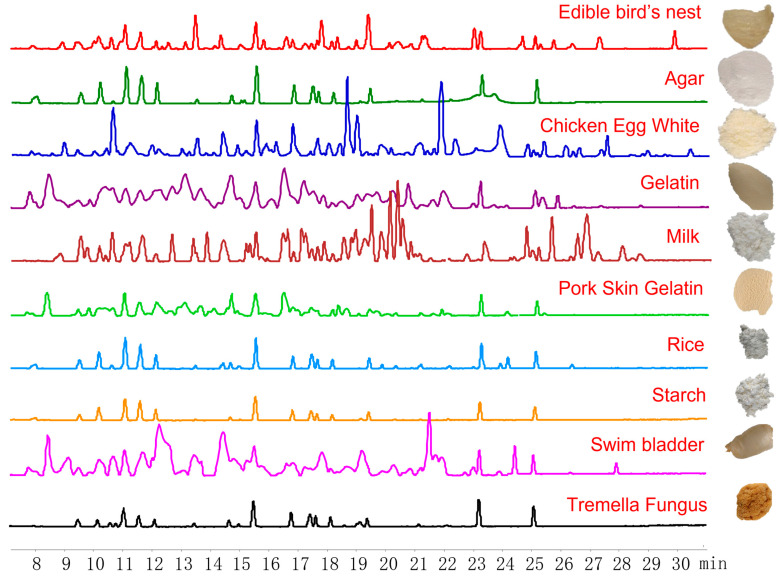
Base peak chromatograms (**left**) and typical morphology (**right**) of Edible bird’s nest and related adulterants.

**Figure 2 molecules-27-02945-f002:**
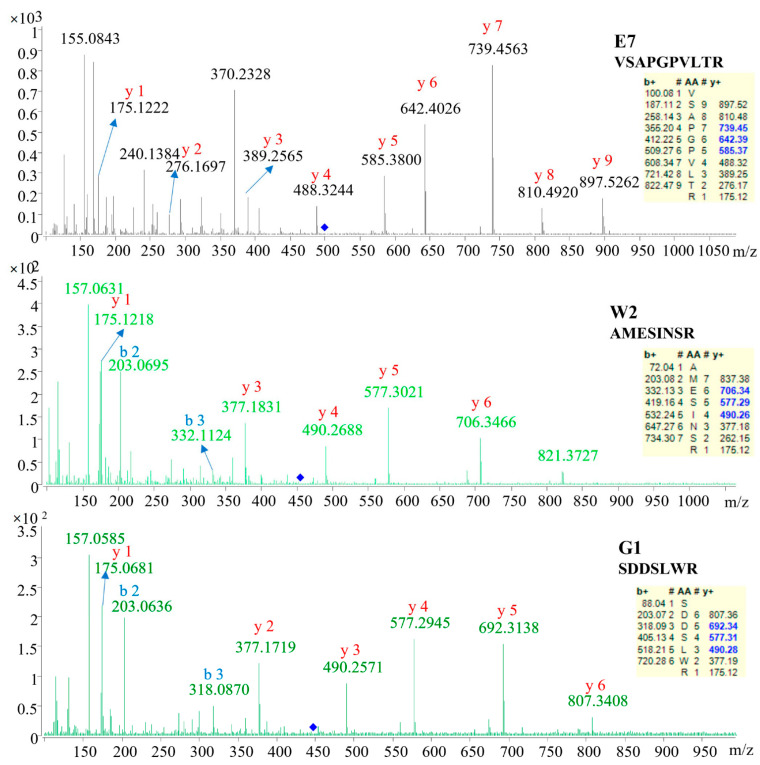
The LC-ESI-MS/MS spectra of ions corresponding to the identified peptide markers.

**Figure 3 molecules-27-02945-f003:**
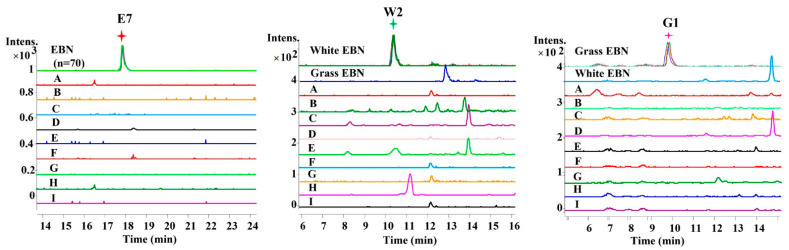
Extracted ion chromatograms (EIC) of selected peptide markers specific to EBN in a typical batch of digested EBN (overlapped) and related adulterants. The adulterants include agar (A), chicken egg white (B), gelatin (C), cow milk (D), pork skin gelatin (E), rice (F), starch (G), swim bladder (H) and tremella fungus (I).

**Figure 4 molecules-27-02945-f004:**
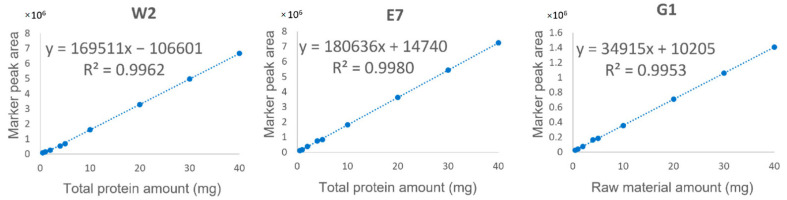
Linearity relationship between the total protein amount/raw material amount and the peak area of specific peptide markers including white EBN specific marker 2 (W2, *m/z* 454.7081), EBN-specific marker 7 (E7, *m/z* 498.8056) and grass EBN-specific marker 1 (G1, *m/z* 447.6998).

**Figure 5 molecules-27-02945-f005:**
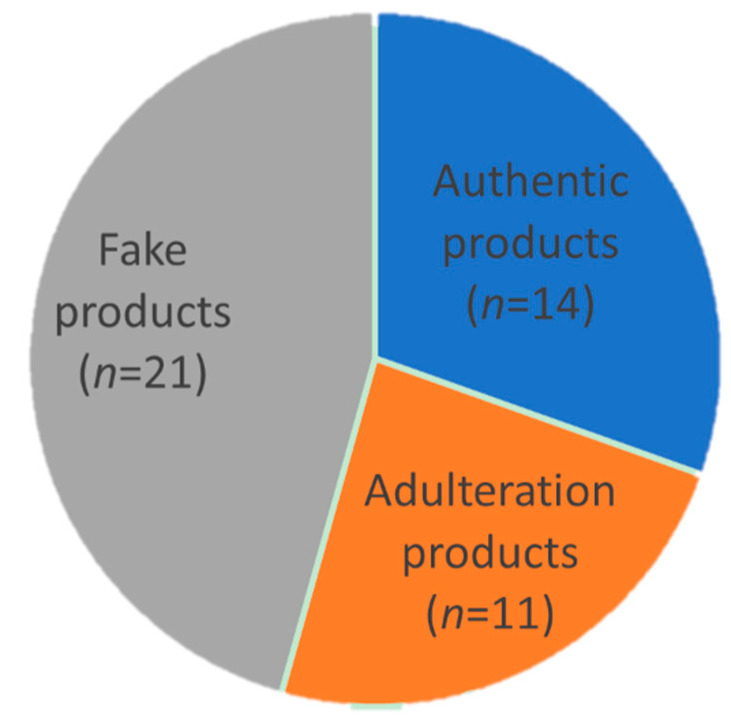
Quantitative analysis result of 46 batches of commercial EBN products. “Authentic products” indicates products which only contain white EBN, “Adulteration products” indicates products containing grass EBN, and “Fake products” indicates products which do not contain EBN.

**Table 1 molecules-27-02945-t001:** Selection of peptide markers of EBN based on retention time, mass-to-charge ratio and specificity.

-	Observed *m/z*	RT (min)	White EBN ^b^	Grass EBN ^b^	Adulterants ^a^	Peptide	Common Peaks ^b^	EBN Peptide Marker
1	375.2250	7.96	+ ^c^	− ^c^	+	−	/^c^	/
2	280.6746	7.97	+	+	−	+	+	E1 ^d^
3	237.1235	9.04	+	−	+	−	/	/
4	272.1722	9.51	+	−	+	−	/	/
5	381.7042	9.61	+	+	−	+	+	E2
6	447.6996	9.81	−	+	−	+	+	G1 ^d^
7	373.8247	9.82	+	+	−	+	+	E3
8	292.1648	10.05	+	−	−	+	+	W1 ^d^
9	548.2486	10.07	+	−	−	−	/	/
10	258.1691	10.24	+	−	+	−	/	/
11	402.2467	10.24	+	−	+	−	/	/
12	454.7018	10.24	+	−	−	+	+	W2
13	467.2138	10.38	+	−	−	−	/	/
14	231.1710	10.67	+	−	+	−	/	/
15	305.1666	10.99	+	−	−	+	+	W3
16	339.1850	10.99	+	−	+	−	/	/
17	582.2528	11.01	+	−	+	−	/	/
18	389.2400	11.09	+	−	+	−	/	/
19	277.6512	11.49	+	−	−	+	+	W4
20	347.6992	11.63	+	−	+	−	/	/
21	382.1620	12.01	+	−	−	−	/	/
22	629.3627	12.56	+	−	+	−	/	/
23	258.6615	12.59	+	−	−	−	/	/
24	288.2036	12.59	+	−	−	−	/	/
25	280.1954	12.86	+	−	−	+	+	W5
26	265.1557	13.54	+	−	+	−	/	/
27	494.2616	14.44	+	−	+	−	/	/
28	350.1712	15.09	+	−	−	−	/	/
29	820.3634	15.13	+	+	−	+	+	E4
30	335.2230	15.39	−	+	−	+	+	G2
31	509.2615	15.79	+	−	+	−	/	/
32	417.7010	16.15	−	+	−	+	+	G3
33	804.3734	16.24	+	−	+	−	/	/
34	404.1986	16.65	+	+	−	+	+	E5
35	366.2030	16.66	+	−	−	−	/	/
36	300.1459	17.25	+	+	−	+	+	E6
37	498.7967	17.87	+	+	−	+	+	E7
38	507.2456	18.39	+	−	+	−	/	/
39	346.2350	19.48	+	−	−	−	/	/
40	364.2235	19.48	+	−	−	−	/	/
41	321.7031	20.45	+	−	−	+	+	W6
42	844.3728	20.63	+	+	−	+	+	E8
43	441.7166	20.65	+	−	−	+	+	W7
44	294.8142	20.7	+	−	−	+	+	W8
45	468.7439	20.78	+	−	−	+	+	W9
46	350.2074	21.07	+	−	−	−	/	/
47	344.255	21.32	+	−	+	−	/	/
48	336.1926	21.4	+	−	−	−	/	/
49	525.7603	22.46	+	−	−	−	/	/
50	477.7330	23.14	+	+	−	+	+	E9
51	630.8100	24.56	+	+	−	+	+	E10
52	479.2879	24.73	+	−	+	−	/	/
53	472.3140	25.78	+	−	+	+	−	/
54	711.4410	26.43	+	−	+	+	+	W10
55	391.7429	27.35	+	−	−	+	+	W11
56	410.7169	27.35	+	−	−	+	+	W12
57	433.7006	28.04	+	−	−	+	+	W13
58	491.3200	29.94	+	−	+	−	/	/

^a^ The adulterants include chicken egg white, agar, gelatin, cow milk, pork skin gelatin, tremella fungus, rice, starch, and swim bladder. ^b^ Common peaks are peptide markers that were consistently present in either 60 batches of white EBN or 10 batches of grass EBN. ^c^ “+” is the positive result. “−” is the negative result. “/” is result that is not applicable. ^d^ “E” is EBN-specific. “W” is white EBN-specific. “G” is grass EBN-specific.

**Table 2 molecules-27-02945-t002:** The peptide markers identified by database searching and de novo sequencing.

Peptide Marker	Observed *m/z*	Retention Time (min)	Charge State	Sequence	Source
E7	498.8056	17.91	2	VSAPGPVLTR	De novo
W2	454.6704	10.37	2	AMESINSR	Database
G1	447.6998	9.88	2	SDDSLWR	De novo

**Table 3 molecules-27-02945-t003:** Validation of the established quantitative method.

Marker No.	Linear Regression ^a^	Range	*R* ^2^	LOD ^b^	LOQ ^b^	Repeatability		Recovery % (RSD, *n* = 3)		
		(mg)		(mg)	(mg)	Intra-Day (*n* = 6)	Inter-Day (*n* = 3)	Low ^c^ (Spiking 80%)	Middle ^c^ (Spiking 100%)	High ^c^ (Spiking 120%)
W2	*y* = 169511*x* − 106601	0.5–40	0.9962	0.05	0.16	1.1%	1.3%	113.6 (4.3%)	111.7 (5.1%)	111.4 (3.8%)
E7	*y* = 180636*x* + 14740	0.5–40	0.9980	0.04	0.12	1.8%	4.7%	117.7 (2.6%)	116.1 (4.5%)	100.6 (5.3%)
G1	*y* = 34915*x* + 10205	0.5–40	0.9953	0.13	0.42	1.1%	3.0%	111.6 (3.3%)	111.1 (6.2%)	105.0 (1.9%)

^a^ The regression of W2/E7 was plotted by total protein (Appendix A) amount vs. marker peak area, while the regression of G1 was plotted by grass EBN raw material amount vs. marker peak area. ^b^ Limits of detection (LOD) of solution were calculated using the definition signal-to-noise ratios (S/N) ≥ 3, while limits of quantitation (LOQ) were calculated using the definition S/N ≥ 10. The result is based on specific markers including white EBN-specific marker 2 (W2, *m/z* 454.7081), EBN-specific marker 7 (E7, *m/z* 498.8056) and grass EBN-specific marker 1 (G1, *m/z* 447.6998). ^c^ The spike recovery test was measured by spiking different amounts (120% high, 100% middle and 80% low level) of total protein amount/raw material amount.

**Table 4 molecules-27-02945-t004:** Quantitative analysis of EBN, white EBN, and grass EBN in commercial EBN liquid products.

Sample Code	W2 ^a^ (mg/mL)	E7 ^a^ (mg/mL)	G1 ^a^ (mg/mL)	Type	Origin
PD-01	12.72 ± 0.60	14.16 ± 0.84	4.60 ± 0.36	Instant product	China
PD-02	3.21 ± 0.04	3.52 ± 0.24	3.40 ± 0.28	Instant product	China
PD-03	7.64 ± 0.12	7.32 ± 0.32	7.52 ± 0.24	Instant product	China
PD-04	12.24 ± 0.48	13.12 ± 0.64	ND ^b^	Instant product	China
PD-05	15.52 ± 0.61	15.60 ± 0.72	ND	Instant product	China
PD-06	5.92 ± 0.28	6.04 ± 0.36	5.41 ± 0.20	Instant product	China
PD-07	52.72 ± 0.92	55.32 ± 2.24	8.96 ± 0.44	Instant product	China
PD-08	61.08 ± 1.56	61.56 ± 2.56	ND	Instant product	China
PD-09	74.80 ± 1.81	75.84 ± 2.96	ND	Instant product	China
PD-10	41.84 ± 1.28	42.68 ± 1.24	5.41 ± 0.36	Instant product	China
PD-11	18.32 ± 0.76	21.88 ± 0.88	10.52 ± 0.48	Instant product	China
PD-12	ND ^b^	ND ^b^	ND	Instant product	China
PD-13	ND	ND	ND	Instant product	China
PD-14	ND	ND	ND	Instant product	China
PD-15	ND	ND	ND	Instant product	China
PD-16	ND	ND	ND	Instant product	China
PD-17	ND	ND	ND	Instant product	China
PD-18	ND	ND	ND	Instant product	China
PD-19	3.20 ± 0.08	3.56 ± 0.09	ND	Beverage	China
PD-20	ND	ND	ND	Beverage	China
PD-21	ND	ND	ND	Beverage	China
PD-22	ND	ND	ND	Instant product	China
PD-23	ND	ND	ND	Instant product	China
PD-24	ND	ND	ND	Instant product	China
PD-25	ND	ND	ND	Instant product	China
PD-26	ND	ND	ND	Instant product	China
PD-27	ND	ND	ND	Instant product	China
PD-28	ND	ND	ND	Instant product	China
PD-29	ND	ND	ND	Beverage	China
PD-30	ND	ND	ND	Beverage	China
PD-31	ND	ND	ND	Instant product	China
PD-32	4.81 ± 0.27	5.52 ± 0.33	ND	Instant product	China
PD-33	ND	ND	ND	Instant product	China
PD-34	22.24 ± 1.12	24.60 ± 1.32	6.64 ± 0.24	Instant product	China
PD-35	18.40 ± 1.16	20.12 ± 1.18	ND	Instant product	China
PD-36	23.92 ± 1.20	24.64 ± 1.39	ND	Instant product	China
PD-37	4.04 ± 0.12	7.01 ± 0.24	12.08 ± 0.44	Instant product	China
PD-38	16.96 ± 0.52	20.68 ± 0.64	ND	Instant product	China
PD-39	8.04 ± 0.24	10.01 ± 0.42	3.32 ± 0.13	Instant product	China
PD-40	71.80 ± 1.88	73.56 ± 2.52	12.44 ± 0.68	Instant product	China
PD-41	14.92 ± 0.64	15.16 ± 0.81	ND	Instant product	China
PD-42	ND	ND	ND	Beverage	Vietnam
PD-43	4.04 ± 0.12	4.24 ± 0.21	ND	Beverage	Vietnam
PD-44	15.44 ± 0.60	15.52 ± 0.68	ND	Beverage	Vietnam
PD-45	13.12 ± 0.44	13.20 ± 0.65	ND	Beverage	Vietnam
PD-46	3.48 ± 0.13	3.52 ± 0.14	ND	Beverage	Vietnam

^a^ The additive amount is calculated by detecting 250 μL EBN products. The content is calculated based on specific markers including white EBN-specific marker 2 (W2, *m/z* 454.7081), EBN-specific marker 7 (E7, *m/z* 498.8056) and grass EBN-specific marker 1 (G1, *m/z* 447.6998). ^b^ “ND” means ”none detected”, which indicates the corresponding value is less than the limits of detection (LOD).

## Data Availability

Not applicable.

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
