# Peer review of "Qualitative and Quantitative Analysis of Edible Bird’s Nest Based on Peptide Markers by LC-QTOF-MS/MS"

_molecules, 2022, doi:10.3390/molecules27092945_

Round 1

Reviewer 1 Report

Title: Qualitative and quantitative analysis of Edible Bird's Nest 2 based on peptide markers by LC-QTOF-MS/MS.
In this research, this work find peptide markers for authentication of EBN, especially white EBN. To this end, the authors developed a simple and effective method for sample treatment, so it avoids time-consuming to analyze large numbers of samples and avoids losing potential markers during the process. Meanwhile, all chemical components, including both soluble and insoluble fraction of EBN, were considered to find peptide markers. Subsequently, a strategy without relying on protein databases was employed to find peptide markers as many as possible. The authors found twenty-six new specific markers. There is a good linear relationship between the peak area of three high signal abundance markers and the total protein content in EBN/mass of EBN, which suggests that they are acceptable for quantitative analysis. The peptide sequences of them are confidently identified by searching database/de novo sequencing. These new markers were successfully applied to detect commercial EBN products.
Overall, the study is well-designed and well-written, the methods are suitable and the obtained results are clearly presented and discussed.  Moreover, the conclusions have been appropriately pointed out.
The content of this manuscript matches well with the journal’s purpose, but some clarifications are you can do:
- table 2: 'm/z' should be italic

- table 3 and 4 also 'm/z' should be italic (in the legend)

 all biography you should correct.

Reviewer 2 Report

General remarks

The manuscript deals with an issue of high interest and importance, from both theoretical and practical sides. However, declaring that the authors find anything “novel” sounds to be too bold when citing only 16 literature references. Edible bird’s nest has a much more extensive literature now, as does the food adulteration issue itself. The below papers are just a few examples from the near past for EBN studies. Please, revise and amend the Introduction part, include a much more extensive literature overview.

  • Tong SR, Lee TH, Cheong SK, Lim YM (2020) Untargeted metabolite profiling on the water-soluble metabolites of edible bird's nest through liquid chromatography-mass spectrometry, Veterinary World, 13(2): 304-316.
  • Mohamad Nasir, N.N.; Mohamad Ibrahim, R.; Abu Bakar, M.Z.; Mahmud, R.; Ab Razak, N.A.: Characterization and Extraction Influence Protein Profiling of Edible Bird’s Nest. Foods 2021, 10, 2248.
  • Ghassem, M.; Arihara, K.; Mohammadi, S.; Sani, N. A.; Babji, A. S. Identification of Two Novel Antioxidant Peptides from Edible Bird’s Nest (Aerodramus Fuciphagus) Protein Hydrolysates. Food Funct. 2017, 8, 2046–2052.
  • Hui-Yuan Cheng, Li-Feng Li, Wen-Jie Wu, Quan-Wei Zhang, Man Liu, Tin-Long Wong, Hau-Yee Kong, Cheuk-Hei Lai, Wan-Rong Bao, Chu-Ying Huo, Hong-Ming Zheng, Qiu-Ke Hou, Jun Xu, Yan Zhou, Quan-Bin Han: Qualitative and quantitative analysis of agar in edible bird's nest and related products based on a daughter oligosaccharide-marker approach using LC-QTOF-MS, Food Control, Volume 132, 2022, 108514,
  • Tan Hui Yan, Sue Lian Mun, Jia Lin Lee, Seng Joe Lim, Nur Aliah Daud, Abdul Salam Babji & Shahrul Razid Sarbini (2022) Bioactive sialylated-mucin (SiaMuc) glycopeptide produced from enzymatic hydrolysis of edible swiftlet’s nest (ESN): degree of hydrolysis, nutritional bioavailability, and physicochemical characteristics, International Journal of Food Properties, 25:1, 252-277
  • Cao, J.; Xiong, N.; Zhang, Y.; Dai, Y.;Wang, Y.; Lu, L.; Jiang, L. Using RSM for Optimum of Optimum Production of Peptides from Edible Bird’s Nest By-Product and Characterization of Its Antioxidant’s Properties. Foods 2022, 11, 859.

Specific remarks

Page 1, rows 40-42. Introduction, paragraph 2:

Various EBN adulteration with less expensive materials, such as agarose and tremella fungus are polysaccharides while gelatin and pork skins are polypeptides, has risen vigorously over the past few years.

One can understand what the authors aimed to express however, this sentence is rather badly formulated in that way.

Page 2, rows 46-47, Introduction.

In addition, the grass EBN is always used as the 46 white EBN instead due to the lack of effective identification approaches.

The same remark.

Page 2, rows 80-81, Results.

So, all molecules, especially proteins, were analyzed to find peptide markers.

Did the authors exclude experimentally that degradation of some components have not occurred? If yes, then please, provide details! If not, then it is not correct to declare that all molecules of the sample were analysed.

Page 3, Table 1. This is hardly decipherable in that way. Please, consider some other way (maybe graphics?) to communicate these results!

The same remark for Table 4 at page 7.

Page 8, rows 185-194, Materials and methods section (and afterwards where applicable): please, put the numbers in chemical formulae to subscript, as it is correct by the IUPAC rules!

Page 8, rows 196-197

[…] was dissolved in 500 μL1% 196 ammonium bicarbonate (ABC) […]

There is a space missing between the characters “L” and “1”.

(Please, check the whole document thoroughly for this kind of spelling mistakes.)

Page 8, rows 206-207

The separation was performed on an […] system […] equipped with a […] thermostatic column […]

Do the authors mean a column thermostat, or – with other words – column oven? Please, clarify this.

Page 9, Conclusions

This is not a concluding section, but rather a second summary. Please, try to draw conclusions about your trial experiences, at least about the analysis of commercial products. Results of these should be better discussed anyway.

Reviewer 3 Report

- Table 1,  the results should be sorted according to the retention time.

- the chapter “Quantitative analysis based on EBN-specific markers” requires revision as required by the validation of the analytical procedure. Statistics analysis for the measurements value are missing.  Calibration curves should be presented. In table 3: the units (ug) LOD and LOQ should be corrected; concentration range of quantitative analysis should be added; the low middle high values should be specified.

In table 4, values of 0.00 should be corrected to

Reviewer 4 Report

General comments:

The manuscript by Wen-Jie Wu et al. number ID: molecules-1648320 “Qualitative and quantitative analysis of Edible Bird's Nest based on peptide markers by LC-QTOF-MS/MS”, developed for the first time a LC-MS/MS method using peptide biomarkers for the discrimination of Edible bird’s nest (EBN) from other adulterants in food products from the market. In my opinion it is a good-quality manuscript. I do not have any major concerns about the study. The manuscript is well organized and written, for these reasons, I recommend its acceptance for publication in Molecules, once the authors have dealt with some minor points detailed below.

Specific comments:

1.- Please indicate the amount of peptides injected in the mass spectrometer per run (?µg).

2.- Please indicate how was performed the MS data analysis. Which software and databases were used.

3.- Please indicate how was developed the de novo sequencing of the peptides. Was performed manually or using a program. Which software were used (i.e. Peaks) and the parameters.

4.- Please improve the quality of Figure 2.

5.- Please improve the quality of Figure 3.

Round 2

Reviewer 2 Report

The manuscript has been improved accordingly; now it is suitable for being published in Molecules.

Author Response

The manuscript has been improved accordingly; now it is suitable for being published in Molecules.

  Authors’ Response------ Thank you very much for your positive comments!

Reviewer 3 Report

The work deals with the interesting problem of identifying the authenticity of Edible Bird's. The authors have already introduced a lot of corrections to the work in relation to the preliminary version. Unfortunately, the analytical part cannot be published in its current form. In table 3 [mg] and table 4 [mg /250uL]  are used. Table 3, calibration curves and table 4 should clearly show how many [mg] EBN proteins (based on peptide markers) are in the tested material [mg of proteins present in the sample or in mg of the sample]. Table 4 does not include statistics on the performed measurements (e.g. standard deviation).
